# Behavioral engagement facilitates auditory neuron responses beyond their receptive fields

**Chenggang Chen**[‡], **Evan D. Remington**[‡], **Xiaoqin Wang**[*]

Department of Biomedical Engineering, Johns Hopkins University School of Medicine, Baltimore, Maryland, United States of America

‡ These authors are the co-first authors on this work.
* xiaoqin.wang@jhu.edu

## Abstract

In the auditory cortex, neural responses to stimuli inside receptive fields (RFs) can be further facilitated by behavioral demands, such as attending to a spatial location. It is less clear how off-RF stimuli modulate neural responses and contribute to behavioral tasks. Our recent study revealed a particular form of location-specific facilitation evoked by repeated stimulation from an off-RF location, suggesting behavioral modulation of spatial RFs. To further explore this question, we trained marmosets to attend to sound locations that were either inside or outside the RFs of auditory cortical neurons. The majority of neurons showed increased firing rates at target locations inside their RFs. Interestingly, this increase also occurred outside the RFs, sometimes exceeding the responses at the RF center during passive listening. This task-related off-RF facilitation was much more common in the caudal area than in the rostral area and the primary auditory cortex. A normalization model reproduced the off-RF facilitation using widespread suppression. The model's prediction was confirmed by experimental observations of widespread reductions in firing rate and hyperpolarized membrane potentials for off-RF stimuli. These results suggest that behavioral task demands recruit a broader range of neurons than those that are responsive to a target sound in the passive state.

## Introduction

It is well known that response properties of neurons in the visual [1–4], somatosensory [5–7], and auditory [8–13] cortices can change rapidly to match specific task demands. In the visual cortex, this phenomenon has been studied extensively for visual location, for which attending to behaviorally relevant locations inside receptive fields (RFs) increases neural activity [14–21] and reduces the inter-neuronal correlations [22–24]. Similarly, attending to different locations of somatosensory stimuli inside the RFs of the finger or whisker also facilitates neural responses [25,26]. In the auditory cortex, extensive work on frequency tuning shows sharpened RFs [27–32]

**Data availability statement:** All relevant data are within the paper and its Supporting information files. Raw and processed data is available at Zenodo (https://zenodo.org/records/18736680) and GitHub (https://github.com/ccg1988/Attention_PLoS_Biology_2026).

**Funding:** This work was supported by National Institutes of Health (https://www.nidcd.nih.gov/, DC003180 to X.W.). The funders had no role in study design, data collection and analysis, decision to publish, or preparation of the manuscript. C.C., E.R., and X.W. received salary support from the National Institutes of Health.

**Competing interests:** The authors have declared that no competing interests exist.

**Abbreviations:** ANOVA, analysis of variance; AZs, azimuths; FA, false alarm; FM, frequency modulated; LMM, Linear Mixed-Effects Model; MI, modulation index; RFs, receptive fields; RSS, random spectrum stimuli.

and mainly enhanced, occasionally suppressed, responses at target frequencies within the RFs of single neurons or the majority of simultaneously recorded neurons [33–39].

A common procedure for a sensory attention task is to present a target stimulus that falls inside or near the RFs of recorded neurons. Visual or somatosensory location and auditory frequency are faithfully relayed from the sensory receptors to the cortex, with topographic maps preserved across all the stations along the ascending pathways [40]. A stimulus far from the RF is not supposed to evoke a neural response. In contrast, the sound location information is computed from the sensory inputs from the two ears by the auditory system [41,42] and is then used in a variety of behavioral contexts beyond sound localization [43–46], such as sound segregation in a cocktail party [47]. We recently showed that when a sound stimulus was repeatedly presented from a particular location outside the RF of a neuron, the neuron could temporarily increase its responses to this location, suggesting facilitation or attention-based mechanisms [48]. To fully understand such mechanisms, one needs to first sample the entire auditory spatial field beyond the azimuth and elevation axes to determine the extent of the RF of a neuron. However, sampling spatial locations that fall on a three-dimensional sphere (not considering distance) is technically challenging in neurophysiological recordings.

The auditory cortex is essential for many behaviors involving sound localization in mammals [49–55]. Neural responses in the auditory cortex are also modulated by spatial attention. Studies in humans using non-invasive imaging methods [56–66] and surface electrodes [67] all found enhanced responses when attention is directed to preferred sound locations, except for one study which showed similar responses to preferred locations but decreased responses to nonpreferred locations [68].

Animal studies, which have single-neuron resolution, have shown varied effects of task engagement on responses in the auditory cortex. These effects include increased firing rates to sounds played to an attended ear in a selective listening task [69], increased firing rates to arbitrary locations in a localization task [70], and general firing rate increases in an interaural phase discrimination task [71]. In contrast, in an elevation discrimination task, responses to preferred locations were consistent, but responses to nonpreferred locations decreased, resulting in narrower tuning widths [72,63]. In addition, for the spatially untuned neurons in a free navigation task, half of them showed increased responses and the other half showed decreased responses to the target speaker [73]. Together, previous studies suggest that attending to preferred sound locations increases neural activity, but the effect could be facilitative or suppressive at nonpreferred locations.

All of the aforementioned studies measured responses to a subset of spatial locations, including contralateral and ipsilateral locations [56,57,60,62,67,69,74], front azimuth [59,61,66,68,70], azimuth localization cues [58,64,65,71], or the full azimuth plane [63,72,73]. Preferred and nonpreferred locations cannot be accurately determined until the full spatial field (azimuth and elevation) has been sampled. For the first time, we mapped the neural representations of the full spatial fields in the auditory cortex of awake, passively listening subjects [75]. This study revealed a

widespread distribution of preferred locations of auditory cortex neurons, which was not limited to the front space or azimuth plane. The importance of using a larger sampling space was further exemplified by our previous work showing that a masker sound from a nonpreferred location inhibited neural responses to a probe sound from a preferred location [76,77], and our recent work showing neural facilitation to repetitive stimuli from nonpreferred locations outside a neuron's spatial RF [48]. However, little is known about how auditory cortex neurons represent the large or full spatial field [78–82] during active listening.

To address this question, we recorded single-unit responses in the auditory cortex of marmosets (*Callithrix jacchus*), an arboreal New World monkey, while they performed a spatial discrimination task [83]. Unlike earlier auditory attention studies that sampled only a portion of space, we trained marmosets to attend to targets from the entire spatial field. Presenting full-field stimuli enabled us to accurately identify each neuron's spatial RF during a passive state, and then measure task-related modulations in both preferred and nonpreferred locations. We found a task-related facilitation effect for both preferred and nonpreferred locations. Taken together with our recent findings [48], these results suggest that the classical, static view of cortical spatial RFs must be revisited.

This is the first study in a non-human primate species that maps the auditory spatial attention effect across the full spatial field. Our findings demonstrated that top-down attention creates temporary spatial RFs in otherwise unresponsive neurons, revealing a stronger task-modulation effect in the caudal "where" pathway than in the rostral "what" pathway. Using computational modeling, we proposed that widespread spatial suppression in the auditory cortex is a potential candidate circuit mechanism to account for behavioral modulations of spatial RFs, which was validated through novel data analysis, a new stimulus paradigm, and intracellular recordings in awake marmosets.

## Results

We recorded 208 single units from two marmosets while the animals either listened passively to broadband sounds played from speakers on a complete sphere or performed a sound location discrimination task that included a subset of those locations (Fig 1A–1C). Firing rates in the behaving condition were compared to two other conditions: a "passive condition" in which we played sounds in a random order from the full 24-speaker array and used the half-maximum interpolated firing rates as the spatial RFs [75], and a "control condition" in which sounds were played only from target/background locations in the same pattern as the behavioral condition and with roved sound levels. The latter was used to control for the potential confounding effects of stimulus order [48]. In each session, the animals either successfully discriminated ('hit') or failed to discriminate ('miss') the target sound locations. All analyses were conducted on stimuli presented in hit trials, except for the miss trials shown in Fig 4B and 4C. We averaged the firing rates (with spontaneous firing rates subtracted) of all hit or miss trials at each location for each session.

### Attention increased the firing rates at target locations outside the RFs

Fig 1D shows the spatial RF of an example unit, which was computed using the passive responses to 24 randomly presented locations (shown as blue dots in Fig 1E). This unit has a complex RF (black curves), with responses distributed among contralateral bottom and rear locations. In the contralateral front configuration, all four targets and one background were outside the RF. This unit displayed increased firing rates to two of the four targets (#7 and #15), as well as to the background (#8), in the behaving condition (Fig 1F). In the ipsilateral front configuration (Fig 2A), we observed a consistent and even stronger firing rate increase at location #7 (Fig 2B, left).

Furthermore, the spike raster at the target but not the passive location showed a highly reliable and consistent pattern among repeated stimulus presentations (Fig 2B, right). In the contralateral, rear configuration, we observed firing rate increases at all target locations except #3 (S1A Fig). Figs 2C and S1F show a simple RF from another unit. This unit has a near-zero spontaneous firing rate and zero firing rate in 20 locations. However, its firing rates were strongly facilitated at target #23 (Fig 2D, left). Importantly, this unit showed reliable spikes during all five repetitions of the target (Fig 2D, right).

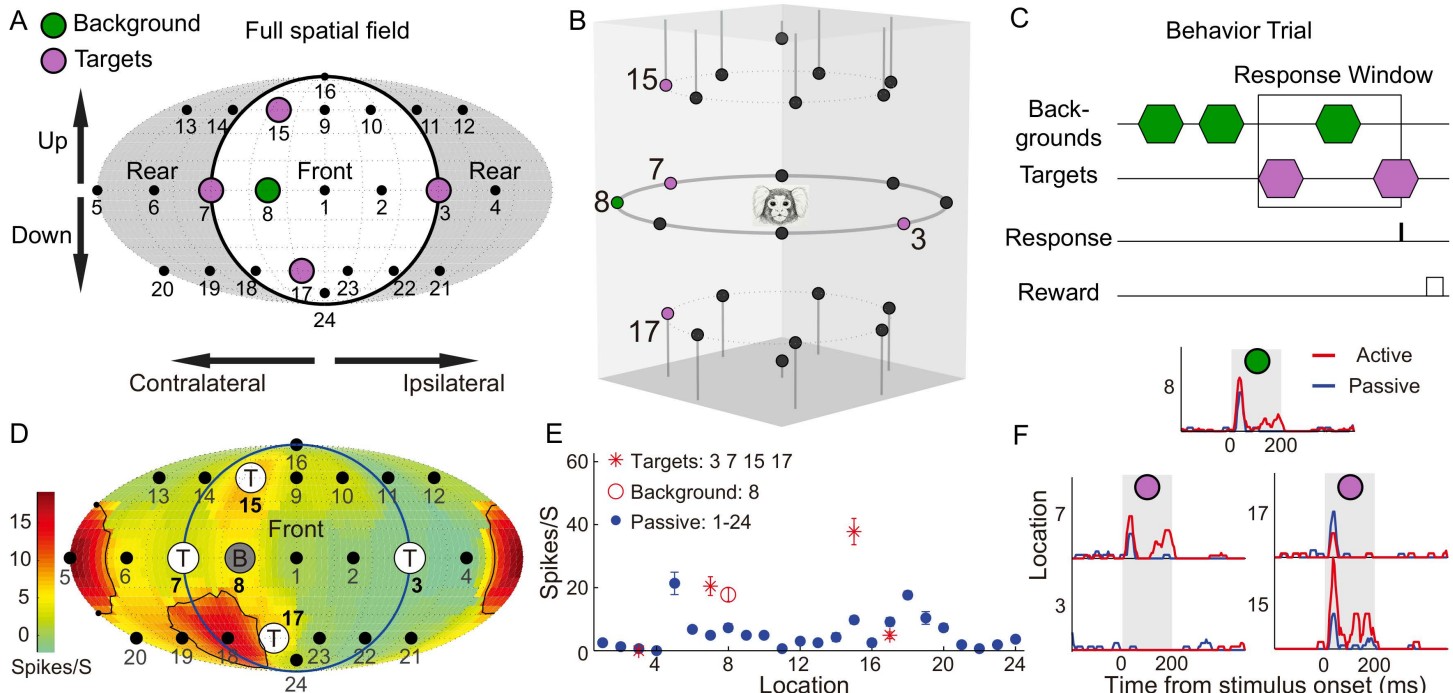

**Fig 1. Full-field sound localization task and example neural responses.** (A) Fournier map projection of a complete 24-speaker array with speakers from (B, C) highlighted. **(B)** Speaker layout for one of the four behavioral conditions. The background location was 45° off the midline, and the target locations were the most lateral positions (±90°; same in all conditions), and also 45° above and below the horizon, but in the same azimuthal quadrant as the background location. Speakers were positioned 1 m from the center of the chamber, which had internal dimensions of 2 × 2 × 2 m. **(C)** After a variable number of background stimuli, targets alternated with the background stimuli. If a lick was registered within the preset number of alternations, a food reward was given. **(D)** Passive spatial RF of an example unit in one animal (M9X-74). Speaker locations, including target and background locations of the discrimination task configuration in **(A)**, were overlaid on the RF. **(E)** Comparison of firing rates measured in the passive (blue) and behaving (red) conditions. Error bars denote the standard error of the mean. **(F)** Peri-stimulus time histogram for four targets (#3, #7, #15, and #17) and one background (#8) location in each condition. The vertical shaded gray bar indicates the 200-ms sound stimulus. Data underlying this Figure can be found in S1 Data.

Compared with the target, we did not find a reliable firing rate increase at the background (S1B Fig). Targets located inside the RFs showed either strongly increased (S1C Fig, #15) or zero (S1E Fig, #7) firing rates compared to the passive condition. All four targets located outside the RFs showed increased firing rates (S1D Fig). S1C–S1F Fig also showed the firing rate changes under the control condition (blue dots and black lines), and we found it was similar to the passive condition but far different from the behaving condition. In summary, comparing targets with passive/control conditions, we observed increased firing rates at one or more specific target locations in the majority of cases.

## Quantification of attention modulations inside and outside the RFs

Fig 3A–3C shows the population distribution of behavioral versus passive comparisons for each target and background location tested for each unit. Population analyses included neurons that displayed a driven firing rate ($p < 0.001$ and minimum mean rate of one spike per stimulus presentation) to at least one location in at least one condition. Although in most studies spontaneous rates have not been observed to change significantly during behavior [8,69], several studies found increased spontaneous rates during behavior [71]. In this sample population, there was a modest but significant increase in spontaneous firing rates when comparing hit trials to passive listening ($p = 3.32 \times 10^{-4}$, Wilcoxon signed-rank test). We quantified the difference using the modulation index (MI), which is a measure of the response difference between two

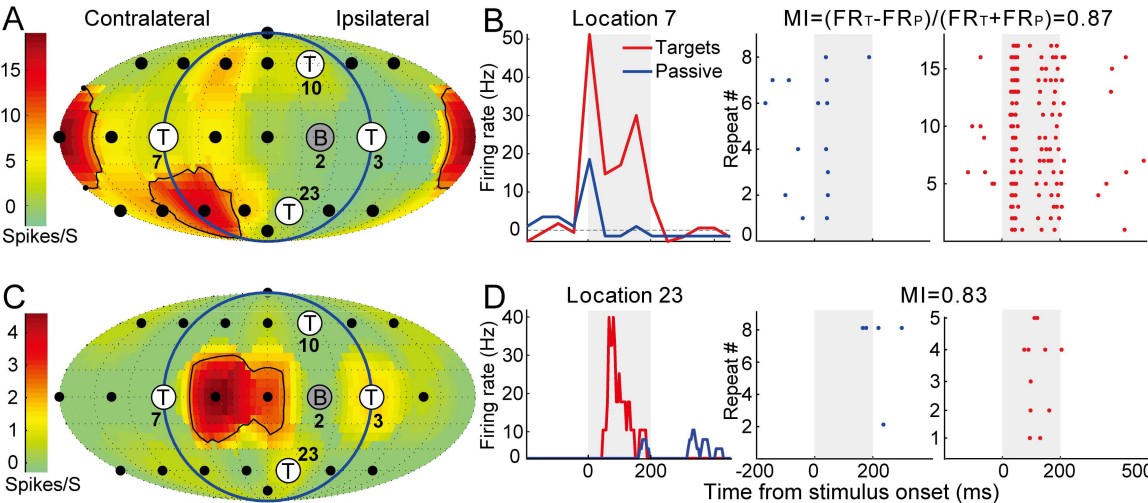

**Fig 2. Firing rates increase at locations outside the RFs during spatial discrimination. (A, B)** Same unit as in Fig 1D–1F but with a different background location (#2) and target locations (#3, #7, #10, and #23). **(A)** Passive spatial RF. **(B)** Left: Peri-stimulus time histogram at location #7. Right: Spike raster in passive (left, 8 trials) and target (right, 17 trials) conditions from location #7. The modulation index (MI) quantifies the firing rate (FR) difference between the two conditions (target minus passive), scaled by the combined responses (target plus passive). **(C)** Passive spatial RF of an example unit in the other animal (M71V-2522). The targets and background locations were the same as in **(A)**. **(D)** Left: Peri-stimulus time histogram at location #23. Right: Spike raster in passive (8 trials) and target (right, 5 trials) conditions from location #23. Data underlying this Figure can be found in S1 Data.

conditions (hits minus passive or control), scaled by the combined responses (hits plus passive or control). MI ranges from −1 to 1, and a positive MI indicates that the firing rate at one location was higher during hits (either at target or background locations) than during the passive or control conditions (Fig 2B and 2D). All following analyses treated each location tested in each neuron as a separate data point.

For target locations that displayed behavioral modulation during hit trials (Fig 3A), we observed elevated firing rates in the majority of locations (83%, purple) during the behavioral condition (Y-axis) compared with the passive condition (X-axis). The MI was 0.48 (median) for all target versus passive pairs. Of the units showing significant modulation, 70 (79%) had at least one significantly increased firing rate, while 19 (21%) had at least one significantly decreased rate. In contrast, for background locations (also in hit trials, Fig 3B), the effects were mixed: we observed both increased and decreased firing rates, with an MI of 0.22. There were 17 (53%) units that had at least one significantly increased firing rate to background locations, and 14 units had at least one significantly decreased firing rate.

Averaging the firing rate in hit trials of one session at one target and background location, and treating each value as an independent data point, may violate the assumption of statistical independence. We addressed this concern with two additional analyses. First, to account for repeated measures from the same units, we employed a linear mixed-effects model [84] with task condition (hit versus passive) and spatial location as fixed effects, and unit identity as a random effect. The model formula was: firing rate ~ condition + location + (1 | unit ID). We found a significant main effect of task condition ($F = 60.8$, $p = 9.3e−15$), confirming that engagement in the sound localization task robustly modulated firing rates in the auditory cortex independently of spatial tuning preferences ($F = 21.2$, $p = 8.2e−35$). Second, we averaged the firing rates in hit trials across all sessions and all target and background locations (S2 Fig). These unit-level metrics were nearly identical to the previous location-level metrics (83% and 56%, Fig 3A and 3B): 82% and 56% of units exhibited increased firing rates during hit trials at target and background locations, respectively. These results are also consistent with the percentage of units with at least one significantly increased firing rate at target and background locations (79%

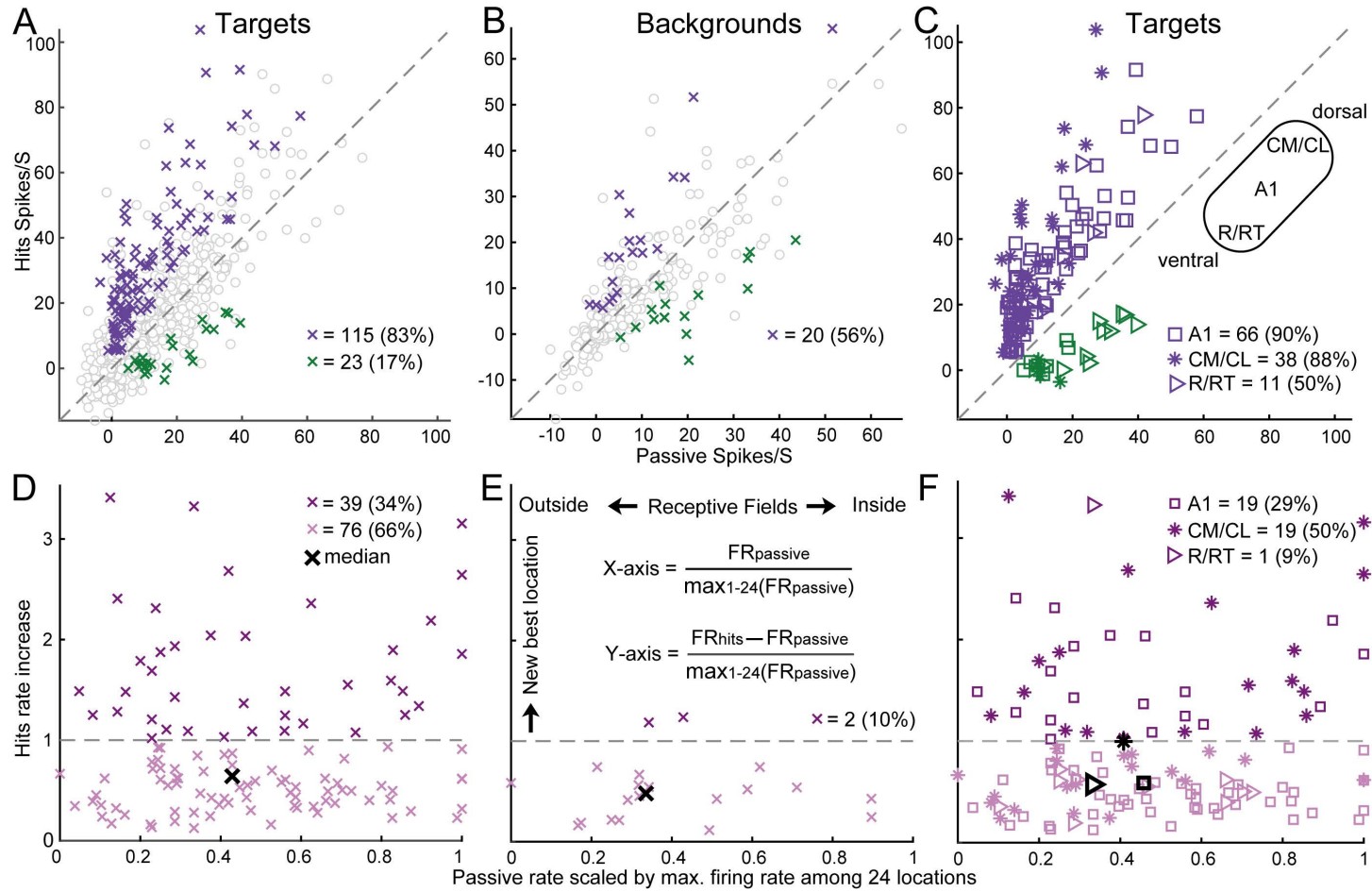

**Fig 3. Attention increased firing rates and generated temporary RFs at target locations. (A)** Comparisons of firing rates in the hit (i.e., successful behavioral choice) and passive conditions at target locations. All firing rates had spontaneous rates subtracted. Colored crosses represent significantly ($p < 0.05$) increased (purple) and decreased (green) firing rates at target locations. Gray circles represent nonsignificant locations (576 (53%) above the diagonal). **(B)** Same as in (A) but for background locations. There were 143 (49%) circles above the diagonal. **(C)** Same data as the colored crosses shown in **(A)**, but with three different shapes (asterisk, square, and right-pointing triangle) to distinguish the three auditory cortical areas (A1, CM/CL, and R/RT). **(D)** Hit firing rate increases relative to the passive firing rate (Y-axis) against the passive firing rate (X-axis) for all increased data points (115 purple crosses in **(A)**), both scaled by the maximum firing rate across 24 locations in the passive condition. The median increase is 0.62 (Y-axis). **(E)** Similar to (D) but for background locations (median: 0.49). **(F)** Same data as in (D) but distinguishing A1, CM/CL, and R/RT (medians: 0.56, 0.98, and 0.55). Data underlying this Figure can be found in S1 Data.

and 53%). Thus, incorporating the mixed-effects model and unit-level metrics provided robust and consistent population-level conclusions.

Therefore, consistent with many previous studies, attending to targets, but not to background locations, increased neural activity. Importantly, as shown in the example neurons (Figs 2 and S1), increases occurred even in neurons with very low passive firing rates (near zero on the X-axis).

Many studies have shown that the distributions of spatial tuning properties vary across auditory areas along the rostral-caudal axis (Fig 3C, inset), with neurons in caudal areas (CM/CL), on average, displaying higher selectivity for spatial locations than those in rostral areas (R/RT) and the primary auditory cortex (A1) [63,75,76,85–87]. However, studies that compared the effects of engagement in a sound localization task among the three areas revealed

conflicting findings. The fMRI studies found that spatial tuning in the human posterior auditory cortex was either task variant [65] or invariant [68]. A neurophysiology study found stronger task modulation in the cat caudal area [63]. We therefore asked whether any behavioral effects would differ quantitatively along the rostral-caudal axis in a primate species. Units were assigned to areas using frequency map gradient reversal points (see Methods), and a significant effect of area on MI was observed when comparing behaving and passive conditions (Kruskal–Wallis ANOVA, $p$ = $4 \times 10^{-4}$; $p$ = 0.03 for A1 versus CM/CL; $p$ = 0.02 for A1 versus R/RT; $p$ = 0.003 for CM/CL versus R/RT, corrected for multiple comparisons), with the largest values found in CM/CL, and the smallest in R/RT (Fig 3C). There were 36 (86%), 24 (83%), and 10 (56%) units with significantly increased responses in A1, CM/CL, and R/RT, respectively. Together, attention primarily increased firing rates in A1 and CM/CL neurons, whereas the effect was mixed in R/RT neurons.

### Attending to target locations shifts spatial response profiles

Firing rate changes during sound localization tasks have been observed to correlate with the underlying spatial RFs [63,69,72], but not in all cases [70]. As discussed in the introduction, these studies measured firing rates at only a sub-set of spatial locations. The interpretation of these effects will benefit from a more complete sampling of spatial RFs, as evidenced by our recent study demonstrating location-specific facilitation across a half of full spatial field [48]. In this study, example units (Figs 2 and S1) and the population analysis (Fig 3A–3C) showed that attention increased firing rates at nonpreferred target locations exhibiting weak or no activity.

Here, we quantified the increases occurred along the passive response tuning curve, defined as the passive firing rate scaled by the maximum passive firing rate across the 24 locations (Fig 3D–3F). On the X-axis, a value of 1 indicates that the target/background location evoked maximum responses from the center of the RF (e.g., #15 in S1C Fig was near the RF center). A value of 0.5 indicates that the location fell near the RF boundary (e.g., #17 and #15 in Fig 2A). A value close to 0 indicates that the location fell outside the RF (e.g., #3 in Fig 2A). Among the 115 target locations that exhibited significantly increased responses during hits compared to passive conditions (purple crosses, Fig 3A), the median scaled passive rate (0.43, black cross, Fig 3D) was close to the RF boundary (i.e., 0.5). Among the 20 background locations (Fig 3B), the median scaled passive rate was 0.33 (Fig 3E). The difference between these two values indicates that during the passive condition, target locations (comprising both azimuth and elevation) evoked larger responses than the azimuth-only background locations.

The Y-axis in Fig 3D–3F represents the magnitude of increase, scaled by the maximum firing rate at the RF center. We found that increases in firing rate occurred throughout the RF, as indicated by the large number of increases at locations outside the RF, which drove the neuron poorly in the passive condition (Fig 3D). Increases did not occur at all locations in units with increased responses, and this effect was not due to the predominance of non-driven locations, which rarely showed firing rate increases. Out of a minimum of 4 locations tested, the median number of significantly increased responses per modulated unit was 1, while the median number of driven locations in the same population of units was 3 ($p$ = $4 \times 10^{-9}$), indicating that increases did not occur uniformly across the RF. The horizontal dashed line (value of 1) indicate where the magnitude of increase exceeded the firing rate at the RF center. Surprisingly, attending to a location temporarily created a new best location (i.e., RF center) for 34% (Fig 3D) of target locations but only 10% (Fig 3E) of background locations.

We further separated three cortical areas for the targets using the same data shown in Fig 3D. Although the percentage of significantly increased locations was similar between A1 and CM/CL (Fig 3C), the percentage of locations above the horizontal line was higher in CM/CL than in A1 (Fig 3F). The median scaled hit firing rate increases were similar between A1 and R/RT, but were much higher in CM/CL (closer to 1). The difference between CM/CL and A1/ R/RT was also significant ($p$ = 0.0013, rank-sum). Together, attention generated new spatial responses outside the RFs mainly in CM/CL neurons.

## Attentional modulation is context-independent and behaviorally relevant

To rule out the possibility that firing rate modulation was attributable to stimulus context, we recorded from a subset of neurons during a 'control' condition in which the stimuli were identical to those in the behavioral condition. In the control comparison, the MI for target locations (0.50; Fig 4A, left) was similar to that of the passive comparison; of 111 units tested, 37 and 2 had at least one significantly increased and decreased firing rate to targets, respectively. The control MI for backgrounds of 0.26 (Fig 4A, middle) was larger than that for the passive comparison but did not reach significance ($p = 0.15$); 16 and 5 units had at least one significantly increased and decreased firing rate to backgrounds, respectively. We further separated significantly modulated locations at target locations into three cortical areas (Fig 4A, right). The differences among three areas were no longer significant compared to the hits/passive condition previously shown (Kruskal–Wallis ANOVA, $p = 0.1$). This lack of effect is likely partially due to the reduced statistical power of the hits/control comparisons ($n = 37$ units compared with 86 for the hits/passive comparison), but also due to the increased MI (0.42) in areas R/RT. This suggests that the stimulus order in the control condition may depress firing rates compared to the random stimulus order in the passive condition, particularly in areas R/RT. Together, the enhanced firing rate during the behavioral condition is not due to stimulus order or temporal predictability.

Given that correctly localized sounds increase the firing rate of auditory cortical neurons (i.e., positive MI), we examined neural responses when the animal failed to localize the target (miss trials). Fig 4B (left) shows that during miss trials, only 63% of target locations exhibited significantly increased firing rates. The median MI was 0.3, which is significantly lower than that of hit trials (0.48; $p = 5.9e−04$, rank-sum test). For background locations (Fig 4B, middle), miss trials tended to decrease the firing rate relative to the passive condition (median MI: −0.15). When examining the three cortical areas separately (Fig 4B, right), the proportion of locations with increased firing rates was substantially lower during misses compared to hits (74%, 68%, and 40% versus 90%, 88%, and 50%). We further directly compared firing rates between hit and miss trials at target locations (Fig 4C) and found that successful localization elicited higher firing rates (78% above diagonal; median MI = 0.28). Thus, the enhanced firing rate observed during behavior is performance-dependent, occurring primarily during hit rather than miss trials.

While miss trials reflect behavioral performance, relying solely on them has limitations; a miss may simply result from low arousal or inattention. A metric that captures both performance and arousal state is the false alarm rate (i.e., licking in the absence of a target) which indicates high arousal or motor readiness. To distinguish between specific auditory processing and general arousal or motor preparation, we analyzed the relationship between neural modulation (MI) and behavioral sensitivity ($d'$), which accounts for both hit and false alarm rates. We divided all significantly modulated locations into groups with positive and negative MIs and visualized the relationship between perceptual sensitivity and decision probability (Fig 4D). If firing rates are positively correlated with task performance (i.e., positive MI aligns with higher d'), the activity likely reflects auditory processing, as the neuron differentiates between a correct detection (hit) and an impulsive error (false alarm) despite similar motor behavior. Conversely, if activity mainly reflects motor preparation or arousal, the neuron would fire whenever the animal licks regardless of accuracy. Our results support the former hypothesis: the median d' of the positive MI group (1.375) was nearly double that of the negative MI group (0.684). Importantly, the median false alarm rate was identical (0.2) for both groups. In summary, enhanced neural responses correlate with task performance rather than non-specific motor preparation or arousal.

## Modeling the dynamics of spatial responses with widespread suppression

What is the neural mechanism underlying this firing rate increase both inside and outside the spatial RFs? To model this auditory spatial attention effect, we chose the widely used divisive normalization model [88]. The normalization models were first developed to model visual responses to stimuli of different contrast [89], and were subsequently used for visual spatial attention [90–92], spectrotemporal contrast [93], and multisensory integration [94]. Importantly, some predictions of the normalization models have been validated experimentally [95–97].

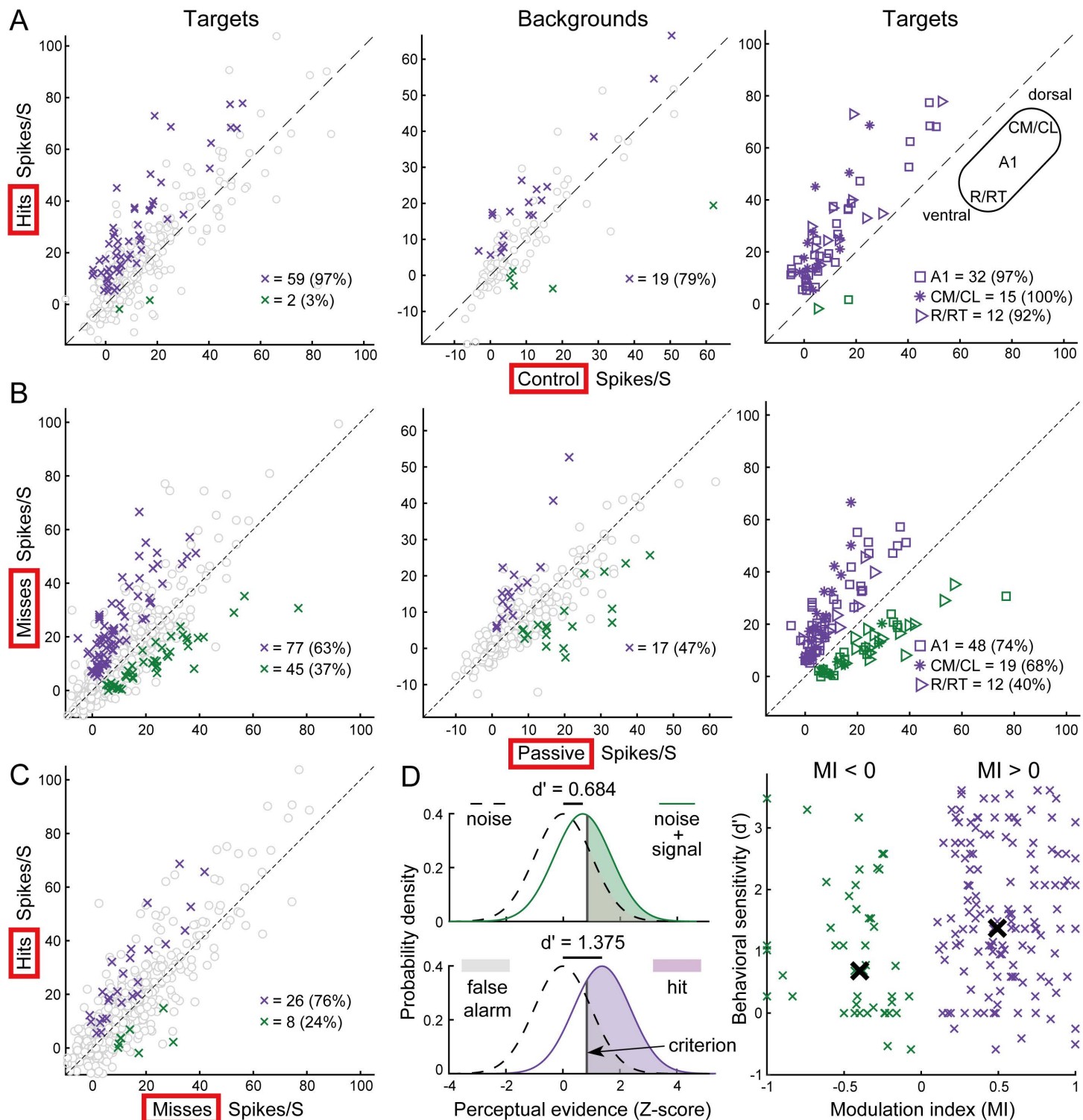

**Fig 4. Context-independent but task-performance-dependent firing rate modulations. (A)** Firing rates for hits vs. control. Left: Target locations. Gray circles indicate non-significant modulation ($n$ = 249 above, 199 below diagonal). Purple and green indicate significantly increased and decreased rates, respectively. Middle: Background locations ($n$ = 53, 55). Right: Significant target locations separated by cortical area (shapes). **(B)** Firing rates for misses vs. passive. Left: Target locations ($n$ = 413, 397). Middle: Background locations ($n$ = 141, 149). Right: Significant locations separated by cortical

area. **(C)** Firing rates for hits vs. misses at target locations (*n* = 435, 300). **(D)** Left: Schematic of signal detection metrics (false alarm, hit rates, criterion) and sensitivity (*d'*). Right: Scatter plot of d' vs. modulation index (MI) for significantly driven targets (*n* = 197). Black crosses indicate medians for positive (*n* = 148) and negative MI groups. Data underlying this Figure can be found in S1 Data.

There are three fields in this model (Fig 5A): stimulus (left), attention (top), and normalization (or suppression, bottom). On each simulated trial (or target location), the stimulus field is multiplied by the attention field and divided (i.e., normalized) by the suppressive field to generate the output firing rate. We modeled all three fields as two-dimensional Gaussian kernels within a 29 × 29 grid. The stimulus field was always centered and had a fixed standard deviation (σ = 2). The attention and suppression fields could drift away from the center, and only the suppression field could broaden (σ was free). Our model recreated the experimental results shown in Fig 3A (Fig 5B, left) and Fig 3D (Fig 5B, right). The three colored crosses (magenta, black, green) highlighted example target locations that show increased, unchanged, and decreased

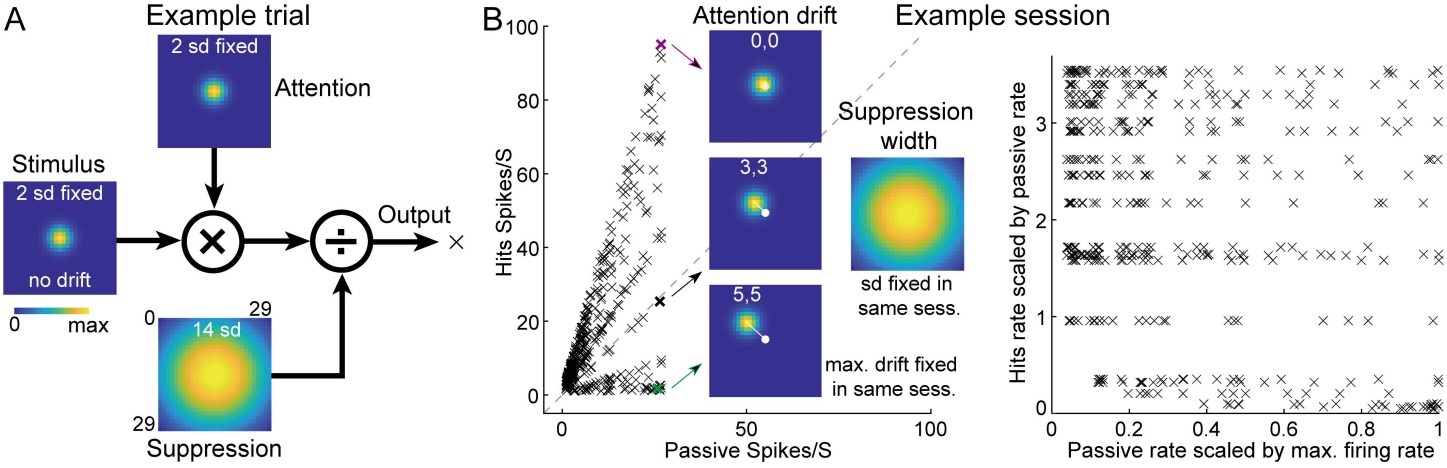

Experimental constraints: proportion of attention-suppressed locations and the ratio of inside- to outside-RF locations

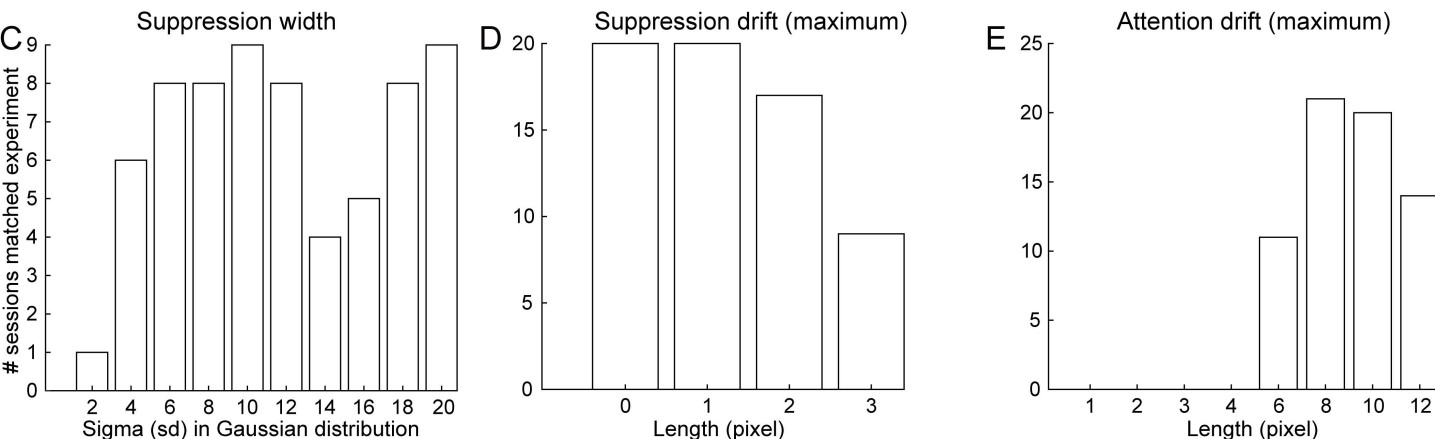

**Fig 5. A normalization model reproduced experimental findings and revealed a candidate neural mechanism. (A)** Schematic of the normalization model showing the three fields, multiplication, and division. **(B)** Left: firing rates during hits and passive conditions. Three example target locations have a fixed suppression width but with different attention drift. Right: hits firing rate increases (relative to passive) plotted against passive firing rate for all data points showing enhancement. **(C–E)** Distributions of model parameters constrained by the experimental results: **(C)** Suppression-field width (σ), **(D)** Maximum suppression-field drift, and **(E)** Maximum allowable attention-field drift. Data underlying this Figure can be found in S1 Data.

firing rates due to attention, respectively. In each session, the standard deviation (or width) of the suppression field and the maximum allowable drift of the attention and suppression fields were fixed. Figs 5C–5E show the histograms of 66 simulated sessions that matched two experimental constraints. One was that the proportion of attention-suppressed locations (points below the diagonal in (B), left) lay between 10% and 30% (experiment: 21%). The other one (S3A Fig) was that the ratio of inside-RF locations (scaled passive rate 0.7–1.0 in (B), right) to outside-RF locations (0.2–0.5) exceeded 40% (experiment: 62%).

In all modeling sessions that matched experimental results, we found that the width of the suppression field had to be larger than the widths of the attention and stimulus fields (Fig 5C). Only one session succeeded with the same width as attention/stimulus fields, whereas many sessions succeeded with a very broad suppression field. The remaining two parameters were the maximum allowable drifts of the suppression and attention fields. Most successful sessions had little or no drift in the suppression field (Fig 5D). The maximum allowed attention drift had to be larger than 6 pixels (Fig 5E). To align with the experimental criteria (at least one spike at a location), we excluded all trials (or locations) with output firing rates below one spike per second. All retained locations ended up having actual attention drifts between 0 and 5 pixels (S3B and S3C Fig). Together, the normalization model recaptured our experimental observations. It further predicted that widespread suppression in the spatial RFs plays a necessary role.

## Widespread suppression is more prominent outside spatial RFs than spectral RFs

Our computational model suggests that widespread suppression in spatial RFs contributes to firing rate increase both inside and outside the RFs. A suppressed firing rate relative to the spontaneous rate at nonpreferred locations has been observed in the auditory cortex of passively listening cats [98] and macaques [87]. Here, we reanalyzed spatial RFs published previously [75] recorded during passive listening (which included marmosets used in this study). Fig 6A shows an example unit with a high spontaneous firing rate (20 spikes/s) that showed suppression at 18 sound locations (blue regions in the RF). Fig 6B further shows that suppression was not limited to ipsilateral space but was widespread across both contralateral and ipsilateral locations. To quantify suppression in the population (Fig 6C, left), we selected 123 units and used a similar analysis to that which we performed previously for spectral tuning (S4A and S4B Fig; [99], their Fig 4).

We also selected 59 units (from the same population used for spatial tuning) following the same criteria used for the spatial tuning (Fig 6C, right). Compared to spectral RFs, both the suppressed area (7.6 versus 1.3) and number of suppressed stimuli (59 versus 28) were much larger in spatial RFs (Fig 6D). The limited suppression observed in spectral RFs was consistent with previous studies (specifically regarding onset and sustained responses; [99]).

To observe suppression below the spontaneous firing rate, neurons must exhibit sufficient spontaneous activity. To drive neurons with low or no spontaneous firing (Fig 7A, left) while measuring spatial RFs simultaneously, we designed a novel "random spatial profile" stimulus that simultaneously delivers white noise from the 24-speaker array at randomly chosen sound levels. This design was inspired by random spectrum stimuli (RSS) [100]. RSS are a class of parametric wideband stimuli capable of driving neurons in multiple cortical fields and have been applied to study auditory spatial tuning [101] and spatial attention tasks [83]. Using this paradigm, we revealed widespread suppression at off-RF locations that showed near-zero firing rates in passively listening marmosets (Fig 7A, right; S5 Fig).

Firing rates only reveal neural responses above the spiking threshold; thus, single-unit recordings cannot reveal suppressive effects below the threshold. Therefore, we used intracellular recordings to measure membrane potentials in response to different locations in the half spatial field in passively listening marmosets [102]. Fig 7B shows the spatial RF (top) and membrane potentials (bottom) for two different locations. Note that the membrane potentials were hyperpolarized (i.e., more negative) after (up arrow) and during (down arrow) sound presentation. In another example unit (Fig 7C), there was no sound-evoked firing since its membrane potentials were suppressed at all 15 sound locations. Combining the novel sound stimuli and intracellular recordings, we were able to reveal widespread suppression at many "unresponsive" sound locations.

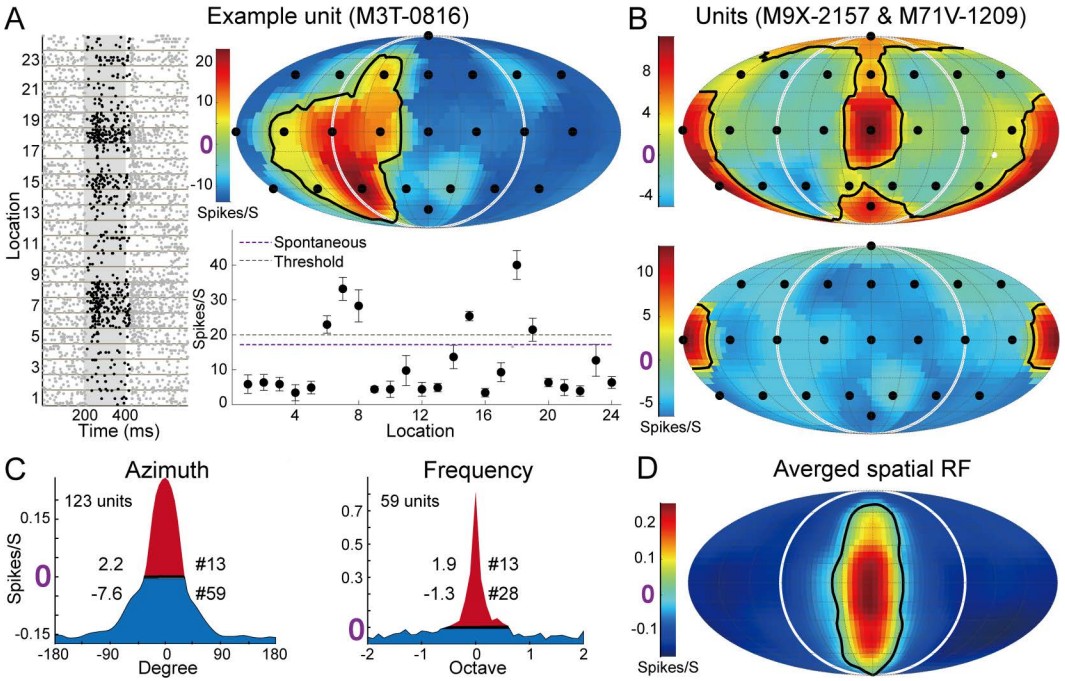

**Fig 6. Firing rate suppression evoked by stimuli from locations outside spatial RFs. (A)** Spike raster (left), spatial RF (top), and average firing and spontaneous rates (bottom) for an example unit preferring the contralateral bottom location. **(B)** Two example units preferring the meridian plane (top) and back (bottom) showing suppressed firing rates at multiple locations (blue). **(C)** Widespread suppression among the population for spatial and spectral RFs. Only units with spontaneous firing rate > 1 spike/s and more suppressed (blue) than excited stimuli (red) were selected. 1-dimensional azimuth (left) and frequency (right) tuning were normalized, circularly shifted, and averaged. Values on the left and right sides of each curve indicate summed areas and stimulus number, respectively. **(D)** 2-dimensional averaged spatial RFs. All data were previously published in a study of passively listening marmosets [75]. Data underlying this Figure can be found in S1 Data.

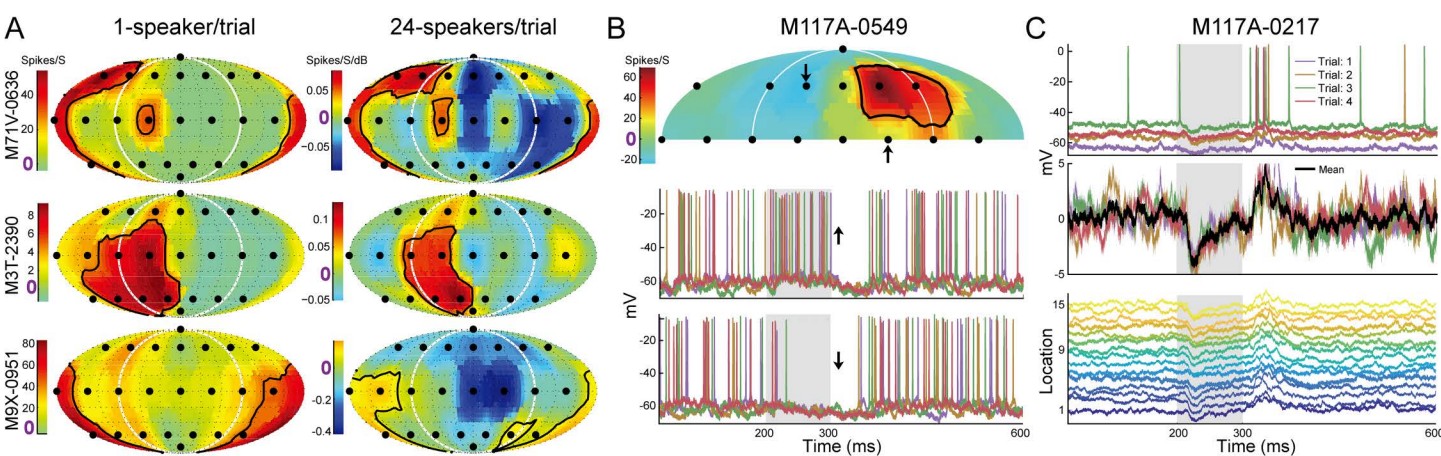

**Fig 7. Firing rate suppression and membrane potential hyperpolarization evoked by stimuli from locations outside spatial RFs. (A)** Spatial RFs of three example units (rows) under two stimulus paradigms (columns). **(B)** Intracellular recordings from the auditory cortex of an awake marmoset. Top: spatial RF calculated from suprathreshold spikes. Bottom: membrane potentials at two locations showing increased and decreased firing rates and membrane potentials, respectively. **(C)** Top: membrane potential at one sound location across four repeats. Middle: raw and averaged membrane potential traces with spikes subtracted. Bottom: mean membrane potential traces across all 15 sound locations. These data were collected from passively listening marmosets but have not been published previously. Data underlying this Figure can be found in S1 Data.

## Discussion

We studied single-unit responses in the auditory cortex of marmosets while they performed a spatial discrimination task in different regions of the full spatial field. Comparing these responses to those measured while marmosets listened passively, a subset of neurons was observed to have increased firing rates to one or more target locations during task engagement. Effects at background locations were mixed. As the task involved a specific stimulus order which was different from that used to measure spatial RFs, we measured responses in an additional passive control condition in which the stimulus order was identical to that of the behavioral condition. Effects were similar, even slightly larger, when comparing rates with the control condition, indicating that increased firing rates were not driven by stimulus order effects in most neurons. Increases occurred both within and outside the classical spatial RF (typically defined as the half-maximal firing rate area). Comparing effects of behavior among rostral (R/RT), caudal (CM/CL), and primary (A1) auditory areas, the largest effects were observed in CM/CL and the smallest in R/RT.

The observation that firing rate increases occurred at some but not all locations, while also being distributed throughout RFs, indicates that effects in this task were stimulus specific. This has been observed previously in one study of spatial behavior [70] in which subjects were required to localize a sound source. In others, effects tended to occur throughout [71] or in preferred or nonpreferred portions of RFs [69,72], indicating neuron-specific effects. Several studies in behaving ferrets have shown consistent stimulus-specific effects on frequency tuning, but not neuron-specific effects [33]. This dichotomy has also been observed in studies of spatial attention in the visual system, with neuron-specific effects including, but not limited to, multiplicative gain [16] and RF sharpening [103], and stimulus-specific effects exemplified by RF shifts [104].

Another dichotomy in observations of the effects of behavioral engagement on firing rates in the auditory cortex is that some studies have observed increased responses, generally for target stimuli [69,70], while other studies have shown decreases, primarily to background stimuli [8,72]. Although differences have been observed due to task structure, such as between appetitive and aversive tasks [35], these differences were all in studies using positive reinforcement. One study was similar to ours in structure yet had dissimilar results: in cats performing a sound elevation discrimination task, responses were often decreased at nonpreferred locations, while effects at preferred locations were mixed [72]. It may be the case that behavioral effects are dependent on specific details of task structure, such as the distributed backgrounds in Lee and Middlebrooks [72] versus the distributed targets here. Another possibility is that there exist species-specific differences in behavioral effects across different auditory areas. An experiment measuring responses in the same task in multiple auditory cortical areas showed increased responses in the posterior auditory field [63].

While firing rate increases occurred throughout RFs, firing rate increases occurred more consistently and strongly at target locations, and increases were likely to be larger where the target/background contrast was positive in the passive condition. This suggests that these effects acted to tailor spatial representation for the specific behavioral task. Previous studies of mammals performing sound location tasks have observed similar, seemingly optimizing changes in spatial tuning. In the first study, neural responses were recorded in macaques performing a dichotic listening task in which they were instructed to respond to sounds played to the left or right ear. In this task, a population of contralateral-preferring neurons responded more strongly to contralateral locations when the target was contralateral [69]. In addition, cats trained to discriminate sound elevation along the complete azimuthal dimension displayed depressed responses at nonpreferred locations [72]. In both cases, changes occurred that were optimized for the task in the context of the underlying spatial (or binaural) tuning. It has been suggested that this is an underlying high-level property of spatial processing in the auditory cortex [72]; our results seem to add support to this hypothesis.

Several studies have shown that the distributions of spatial tuning properties vary quantitatively (but not qualitatively) across auditory areas along the rostral-caudal axis, with neurons in caudal areas, statistically, displaying higher selectivity for spatial locations than those in rostral areas and primary auditory cortex [63,75,85–87]. Differences then might be expected in the way tasks involving sound location affect caudal versus primary and rostral auditory cortex. A study

comparing effects among A1, PAF, and DZ in cats found a lower fraction of neurons in the posterior auditory field and dorsal zone that sharpened tuning compared with A1 [63]. Additionally, neurons in the posterior auditory field were the only population to consistently increase their responses during behavior.

This report represents the first time behavioral effects on spatial responses in primates have been directly compared among multiple auditory areas, including significant recordings from caudal areas, although effects of behavior outside of the primary auditory cortex have been measured previously, primarily in rostral and lateral areas [69,70]. For the most part, quantitative, not qualitative, differences in effects across areas mirrors quantitative, but not qualitative, differences in tuning properties across areas [76,86,87], although the present data do suggest the possibility of multiplicative in addition to additive gain in areas CM/CL. The most dramatic difference in effect size was the apparent lack of consistent behavioral effects in R/RT; however, it is possible that these differences were the result of stimulus order. When firing rates during behavior in R/RT were compared with the control condition with identical stimulus order, increases became apparent, albeit still smaller than in A1 and CM/CL. To our knowledge, no study has specifically compared the effects of stimulus interaction over long time scales across the rostral-caudal axis.

The varied but significant effects of behavioral context on spatial RFs in the auditory cortex suggest that spatial tuning in the passive condition provides only part of the picture of spatial representation in the auditory cortex. These and other behavioral studies may reconcile disparate observations regarding spatial tuning in the auditory cortex. For example, studies of spatial processing in anesthetized animals have observed a preponderance of neurons with very broad spatial RFs, which almost universally increase in size with increasing sound level [79,80,82]. However, in awake animals, spatial RFs tend to be smaller and do not uniformly increase in size with sound level [76,75,87,98]. The observation that large firing rate increases can occur far from the best location may be evidence of broader inputs that are masked in the passive awake condition when compared to the anesthetized state. Conversely, another behavioral task may lead to still more selective tuning compared to the passive state [72]. We therefore believe that the true nature of spatial representation may not be well understood by studying non-behaving subjects, and that a complete picture will require further behavior studies.

## Methods

### Animal preparation and electrophysiological procedures

Experimental procedures were approved by the Institutional Animal Care and Use Committee of the Johns Hopkins University following National Institutes of Health guidelines **(Protocol Number: PR24M383)**. A chronic recording preparation was used to record single-neuron activity in the auditory cortex (left hemisphere) of two female common marmoset monkeys (*Callithrix jacchus*). Both subjects were trained to sit in a custom-designed primate chair and perform a simple auditory detection task [105] and then a spatial discrimination task [83]. After training, two stainless steel headposts were attached to the skull under sterile conditions with the animal deeply anesthetized by isoflurane (0.5%–2.0%, mixed with 50% O2 and 50% nitrous oxide). The headposts served to maintain a stable head orientation of the subject during electrophysiological recordings, although only one post was fixed in this study. To access the auditory cortex, small craniotomies (1.0 or 1.1 mm in diameter) were made in the skull over the superior temporal gyrus to allow for penetration of single electrodes (tungsten electrodes, 2- to 5-MΩ impedance, A-M Systems, Carlsborg, WA) into the brain via a hydraulic microdrive (Trent-Wells, Los Angeles, CA). Single-unit activity was sorted online using template-based spike-sorting (MSD, Alpha Omega Engineering) and analyzed using custom programs written in MATLAB (Mathworks, Natick, MA). We used intracellular recording procedures identical to those in Gao and colleagues [102]. Recordings were made in the auditory cortex through the intact dura with a concentric recording pipette and guide-tube assembly. Sharp recording pipettes were quartz glass pulled on a laser puller (P-2000, Sutter), and guide tubes were borosilicate glass pulled on a conventional puller (P-97, Sutter). The electrode assembly was advanced perpendicular to the cortical surface using a motorized micromanipulator (DMA-1510, Narishige). Electrical signals were amplified (Axoclamp 2B, Molecular Devices), digitized (RX6, Tucker-Davis Technologies), and saved with custom MATLAB code.

## Acoustic stimuli and receptive field characterization

Experiments were conducted in a double-walled sound-attenuating chamber (Industrial Acoustics, IAC, New York) with internal walls, ceiling, and floor lined with ~3-inch acoustic absorption foam (Sonex). Acoustic stimuli were delivered using an array of 24 speakers (FT28D, Dome Tweeter, Fostex) covering a complete sphere. The loudspeakers were mounted at a distance of 1 m from an animal's head and covered 5 elevations (ELs) at 45° spacing and several azimuths (AZs). One speaker was located directly above the animal, 7 speakers each were evenly spaced at ±45° EL (AZ at −45° EL: ±25.7°, ±77.1°, ±128.6° and 180°; AZ at 45° EL: 0°, ±51.4°, ±102.9° and ±154.3°), 8 speakers were evenly positioned at 0° EL (AZ: 0°, ±45°, ±90°, ±135°, 180°), and finally 1 speaker was located at −67.5° EL at 0° AZ. Subjects sat in a wire mesh primate chair mounted onto a single stainless steel bar such that the animal's head was centered in the room. Marmosets were head-fixed for all recordings. In this text, positive AZ angles correspond to speakers ipsilateral to the recording site or to an ipsilateral shift if changes in azimuth were analyzed. During experiments, eye position was not controlled.

Stimuli were generated in MATLAB at a sampling rate of 97.7 kHz using custom software. Digital signals were converted to analog (RX6, 2-channel D/A, Tucker-Davis Technologies), then analog signals were attenuated (PA5 x2, Tucker-Davis Technologies), power amplified (Crown Audio x2), and played through a chosen channel of a power multiplexer (PM2R x2, 16-channels, Tucker-Davis Technologies). Loudspeakers had a relatively flat frequency response curve (±3–7 dB) and minimal spectral variation across speakers (<7 dB re mean) across the range of frequencies of the stimuli used; all large (5–7 dB) spectral deviations occurred in narrow bandwidths near the upper limit of speakers' frequency range (above 28 kHz), above the first spectral notch measured in marmoset head related transfer functions. Neurons were characterized for frequency, intensity, and spatial tuning. For frequency tuning, stimuli consisted of pure tones, band-pass filtered Gaussian noise, Random Spectral Shape (RSS) stimuli [100,106], and occasionally frequency modulated (FM) sweeps. We sampled the frequency axis in 0.1 octave steps, typically over a 4-octave range (2–32 kHz). All firing rates were calculated over a time window beginning 15 ms after stimulus onset and 20 ms after stimulus offset. Best frequency was defined as the frequency that led to the maximum evoked significant firing rate, or for neurons only driven by RSS stimuli, the highest calculated RSS weight. For spatial tuning, stimuli included band-pass filtered unfrozen Gaussian noise, single RSS stimulus tokens, and occasionally FM sweeps. All stimuli used to measure spatial RFs were either band-pass filtered or constructed to have energy between 2 and 32 kHz. When possible, spatial RFs were measured at multiple stimulus intensities. Stimuli were typically 200 ms long with 10 ms cosine ramps and delivered in pseudorandom order, and, except RSS stimulus sets, delivered between 5 and 10 times and averaged to generate tuning functions.

We characterized spatial RFs in a spatially dense acoustic environment by playing sets of broadband sounds from the entire 24-speaker array simultaneously, randomizing the sound level from each speaker. This stimulus delivery paradigm was adapted from a similar method (random spectral shape stimuli) used to study spectral processing in the auditory system [100,106]. The complete set of stimuli comprised a stimulus matrix $\Lambda$ of intensities in which the rows represent individual stimuli and the columns represent the individual speakers. An RSP set is constructed to sample the space of all possible 24-location spatial profiles in such a way that weighting functions can be calculated to describe the spatial tuning to spatially dense stimuli. To do this, the set must be constructed such that the levels of each speaker are statistically independent across all stimuli. This condition is satisfied if the location intensity autocorrelation matrix is equal to the identity matrix:

$$\Lambda^T \Lambda = I_\Lambda I \tag{1}$$

This constraint can only be met for a matrix having more rows than columns. Therefore, the minimum number of stimuli required to construct a linear estimate of the spatial weighting function in this case is 25. The linear spatial weighting function can be calculated from responses to an RSP set using the following equation:

$$\vec{R}_{lin} = R_0 + \Lambda \vec{w} \tag{2}$$

where $\vec{R}_{lin}$ is a column vector of $m$ rate values predicted in response to a set of $m$ different RSP stimuli, $R_0$ is the firing rate to an RSP stimulus with a flat spatial profile, $\Lambda$ is the mean-adjusted intensity matrix, and $\vec{w}$ is the 24-value linear weighting vector. This equation is referred to as the linear synthesis equation. The weighting function is calculated as:

$$\vec{w} = \frac{\Lambda^{\mathsf{T}} \vec{R}}{n\sigma^2} \tag{3}$$

where $\vec{R}$ is the firing rate vector (in spikes/second) to the RSP stimulus set, $n$ is the number of stimuli in the set, and $\sigma^2$ is the variance of the sound levels at each speaker. Weights $\vec{w}$ are expressed in units of spikes/second/dB. This equation is referred to as the analysis equation. Thus, when the intensity is increased at a location with a positive weight function, the firing rate should increase as well.

### Identification of A1, the rostral fields R/RT, and caudal areas CM/CL

In the marmoset, A1 is situated largely ventral to the lateral sulcus on the superior temporal plane and, similar to other primate species, exhibits a low-to-high topographical frequency gradient along the rostral-caudal axis. The boundary between A1 and the rostral field R can be identified by a downward-to-upward frequency gradient reversal along the rostral-caudal axis. Conversely, areas CL and CM can be identified by an abrupt decrease of best frequency at the high-frequency (caudal) border of A1 [107,108]. In this and our previous studies [75], the boundaries between R/RT and A1 and A1 and CM/CL were set by plotting the average best frequency along the rostral-caudal axis, approximately parallel to the lateral sulcus, and setting a boundary at the local minimum and maximum, respectively, between the two areas. We did not separate neurons further into R/RT or CM/CL, although some studies have found differences in spatial selectivity between areas CM and CL in macaques [87,109].

### Spatial discrimination task

We chose to implement a Go/No-Go type task suited for discrimination behavior. Fig 1C illustrates the behavioral paradigm. The objective in a Go/No-Go task is to respond (a lick at the feeding tube) to target sounds to receive a food reward while withholding responses when a target is not presented. Here, each trial was composed of a variable length "intertrial interval" in which sounds were played only from background locations and a fixed length "response interval" during which target and background locations alternated. Intertrial interval length was randomized between approximately three and 10 stimuli, and the response interval included four target/background alternations. Behavioral responses during the intertrial interval resulted in a time-out, and sometimes a puff of air to the base of the tail, followed by a restarting of the intertrial interval. After the intertrial interval ended, target stimuli were alternated with the background sounds during the response interval. Trials ended when the response interval expired or a lick was detected during the response interval. Behavioral responses during this time were reinforced with approximately 0.1–0.2 ml of food reward. If no response was detected, the next intertrial interval began immediately. One third of trials were sham trials in which stimulus location did not change, and no reward was given for behavioral responses. False alarms were measured with sham trials.

Four target/background configurations were used. The background location was 45° lateral to the midline (front and back, contralateral and ipsilateral), and the target locations were the most lateral positions (±90°; same in all conditions), and also 45° above and below the horizon, but in the same azimuthal quadrant as the background location. For targets above and below the background location, their azimuth locations were either 51° or 25.5° lateral to the midline (one of each per condition). A diagram of one such condition is shown in Fig 1A and 1B. Stimuli were 200 ms in length, and the interstimulus interval was approximately 500 ms, resulting in a stimulus onset asynchrony of approximately 700 ms. Sound

level was roved either ±5 dB SPL or ±10 dB SPL to prevent the use of changes in sound level as a perceptual cue. Mean sound level was chosen to be within the flattest portion of each neuron's rate-level function.

## Effects of behavior on spatial responses

All comparisons between conditions were done using "driven" firing rates, equal to the raw firing rate minus the average spontaneous firing rate measured in that condition. We calculated three measures to quantify the difference in firing rate between behaving and passive conditions. MI is a measure of the response difference at a particular location scaled by the combined response strength at that location:

$$MI = \frac{R_B - R_P}{R_B + R_P}$$

(4)

where $R_B$ is the firing rate in the behaving condition, and $R_P$ is the firing rate in the passive condition. The hit rate increase (Y-axis in Fig 3D–3F) is a measure of the response increase at a location relative to the maximum passive firing rate:

$$S_R = \frac{R_B - R_P}{max(R_P)}$$

(5)

where $max(R_P)$ is the maximum firing rate among all locations in the passive spatial RFs.

We compared rates during active behavior to two different passive conditions. The first condition is the standard spatial RF measured by playing sounds from all 24 speaker locations in a randomized order. To control for effects of stimulus order, which can include suppression or facilitation of neural responses [48], we compared firing rates in the behavior condition to a second condition in which stimulus delivery was identical to the behavior condition except that stimuli did not stop if the animal responded. Data were rejected if a response was made during this control condition, although this was rarely the case. Subjects were cued to the beginning of behavior sessions by alternating the house light on and off.

## Statistics, mixed-effects model, and behavioral sensitivity

Wilcoxon rank-sum tests were used to evaluate the population medians when evaluating statistical significance of populations of values, except when testing for deviations from zero mean, in which case t-tests were used. Correlation analyses were based on Spearman's correlation coefficient. All data analyzed in this study were for neurons and locations that displayed a driven firing rate ($p < .001$, minimum one spike per stimulus presentation) in either the behaving or passive condition. Bonferroni-adjusted $p$-values are reported for tests with multiple pairwise comparisons.

To address concerns regarding statistical independence and account for repeated measures from the same neurons, we analyzed population responses using a Linear Mixed-Effects Model (LMM) implemented in MATLAB (fitlme). Data were first reorganized from a wide format into a long format, where each observation represented the firing rate of a single unit at a specific location and task condition. The model defined Firing Rate as the dependent variable, with Task Condition (Active versus Passive) and Sound Location as fixed effects. Unit Identity was included as a random effect (random intercept) to account for baseline variability across individual neurons. The model structure was specified as:

$$\text{Firing Rate} \sim \text{Condition} + \text{Location} + (1 \mid \text{Unit ID})$$

(6)

The statistical significance of the fixed effects was assessed using analysis of variance (ANOVA) on the fitted model. We validated model assumptions by visually inspecting residual plots, which confirmed that residuals were approximately normally distributed and centered at zero. Given the large sample size ($N > 2,400$ observations), the standard LMM is robust to minor deviations from normality, effectively handling the data structure without the need for generalized linear models.

To quantify behavioral performance, we calculated the sensitivity index ($d'$) based on signal detection theory. For each recording session, hit rates and false alarm (FA) rates were calculated for each of the four target sound locations. To prevent undefined values during z-transformation (where hit or FA rates of 0 or 1 result in infinity), we clipped these rates to a minimum of 0.01 and a maximum of 0.99 prior to calculation. The sensitivity index was computed as:

$$d' = Z(\text{hit rate}) - Z(\text{FA rate}) \tag{7}$$

where Z represents the inverse of the standard normal cumulative distribution function (norminv in MATLAB).

We visualized this framework using a standard equal-variance Gaussian model (Fig 4D, left). In this model, the noise distribution (dashed curve) is centered at 0, and the signal-plus-noise distribution (solid curve) is shifted by the value of d'. The decision criterion (vertical bar) was defined such that the area under the noise curve to the right of the criterion matched the observed FA rate (median = 0.2). Perceptual evidence exceeding this criterion results in a behavioral response.

To examine the relationship between neural modulation and behavioral sensitivity, we analyzed 197 spatial locations that elicited significant sound-evoked responses (Fig 4D, right). These were categorized into two groups based on their neural MI: those with task-enhanced responses (Condition MI > 0, purple) and those with suppressed responses (Condition MI < 0, green). We compared the behavioral performance associated with these groups, observing a median hit rate of 0.667 for the enhanced group compared to 0.5 for the suppressed group, while the median FA rate remained constant at 0.2 for both.

## Modeling

Our normalization model was inspired by two previous studies [92,94]. The first paper proposed a classical one-dimensional normalization model:

$$R(x) = A(x) * E(x)/\sigma + s(x) * A(x) * E(x) \tag{8}$$

Which included $R$ (neuron firing rate), $x$ (center of RF, Gaussian distribution), $A$ (attention field), $E$ (stimulus field), ($\alpha$ semi-saturation constant), and $S$ (suppression field). The second paper proposed a two-dimensional normalization model for multisensory integration (we only used one modality) without an attention field:

$$R(x) = E(x, y)^{\wedge} n/\alpha^{\wedge} n + \frac{1}{N} * \sum_{j=1}^{N} E(x)^{\wedge} n \tag{9}$$

where $n$ represents the exponent of the output nonlinearity.

$$E(x, y) = \exp^{\wedge}(-1 * ((x, y) - (\bar{x}, \bar{y}))^{\wedge} 2/2 * \sigma^{\wedge} 2 \tag{10}$$

where $\bar{x}$ and $\bar{y}$ represent the center of the RF. The standard deviation (s.d.) or the width of the RF was determined by $\sigma$. The key difference from the previous model was the normalization part, which was a weighted sum over nearby neurons. Our two-dimensional normalization model was:

$$R(x) = A(x, y)^{\wedge} n * c * E^{\wedge} n/\alpha^{\wedge} n + S(x, y)^{\wedge} n \tag{11}$$

Here, $n$ was fixed at 2 and $\alpha$ at 32. The size of the RF was $29 \times 29$, and the center of the stimulus field $E$ was fixed at the center (15, 15). The centers of the attention field $A$ and suppression field $S$ were allowed to drift from this center. The

random intensity of sensory inputs was controlled by $c$, which ranged from 8.75 + [1–5]. The maximum allowable drift of the attention and suppression fields was [1–4,6,8,10,12] and [0, 1, 2, 3], respectively. The size of the RF σ was fixed for the attention and stimulus fields but ranged between [2,4,6,8,10,12,14,16, 18, 20]. For the passive condition without attention, we removed the attention field A $(x, y)$.

## Supporting information

**S1 Fig. More example units with firing rate increase outside the spatial RFs (related to Fig 2). (A)** An example session (same unit in Fig 2A) where the background location (#6) and two target locations (#13 and #19) were located at the rear of the animal. **(B)** Spike raster from two example units (Fig 2A and 2C) at the background locations (#8 and #2) under passive (left) and target (right) conditions. Notice there were many more trials under the target locations. **(C–F)** Four more example units. Here, we used black, blue, and red colors to represent 24 locations during passive, control, and behaving conditions (five targets/background locations), respectively. Spontaneous firing rates were indicated with colored dashed lines. Data underlying this Figure can be found in S2 Data.
(TIF)

**S2 Fig. Attention increases firing rates at the single-unit level (related to Fig 3A and 3B). (A)** Comparison of firing rates in hit versus passive conditions, averaged across locations and sessions for each unit ($n = 208$). Gray circles represent nonsignificantly modulated units (99 above and 74 below the diagonal). **(B)** Same as in A but for background locations. Gray circles indicate nonsignificantly modulated units (97 above and 101 below the diagonal). Data underlying this Figure can be found in S2 Data.
(TIF)

**S3 Fig. Increased firing rates to target locations increased firing rates relative to background locations (related to Fig 5). (A)** Firing rate during task plotted versus passive firing rate for all data points that were larger than 1, both scaled by the maximum firing rate in the passive condition. The scaled passive rates between 0.2 to 0.5 and 0.7 to 1 were considered as outside and inside of receptive fields, respectively. **(B, C)** The drift of attention fields in all kept trials from two example sessions. The maximum drifts were only 5 pixels in the kept trials, although the allowable drifts were much larger. Data underlying this Figure can be found in S2 Data.
(TIF)

**S4 Fig. Suppressed neural firing rate to the nonpreferred sound stimuli (related to Fig 6). (A)** Averaged neural firing rates to 31 sound frequencies (4 octaves, 8 stimuli per octave) of an example unit. **(B)** The tuning curve was normalized by the maximum firing rate at 16 kHz (the spontaneous firing rate equated to "0"). It was further circularly shifted so that there were 15 frequencies (2 octaves) at both the left and right sides of the peak firing rate. **(C)** An example unit that was suppressed at all 24 sound locations. Notice that the neuron was still significantly tuned to sound locations (ANOVA, $p < 0.01$). **(D)** A 3D view of averaged spatial receptive fields with both color and height at the third axis represented the firing rate. Data underlying this Figure can be found in S2 Data.
(TIF)

**S5 Fig. Suppressed neural firing rate to the random spatial profile (RSP) sound stimuli (related to Fig 7). (A)** An example unit showed consistent spatial receptive fields under two stimulus paradigms. The position and size of white dots indicate the center of receptive fields and their tuning selectivity, respectively. **(B)** Four more example units all showed suppressed firing rates at more than one sound location using the RSP stimuli. **(C)** The scatter plot (X-axis: 1-speaker, Y-axis: 24-speakers) of neural activities at 24 sound locations under two stimulus paradigms. The correlation ($r$) of neural activities was 0.72. **(D)** The histogram of correlation for all 77 units. Data underlying this Figure can be found in S2 Data.
(TIF)

**S1 Data. Includes the data underlying the** Figs 1D–1F; 2A–2D; 3A–3F; 4A–4D; 5B–5E; 6A–6D; **and** 7A–7C. (XLSX)

**S2 Data. Includes the data underlying the** S1A–S1F; S2A, S2B; S3A–S3C; S4A–S4D, **and** S5A–S5D Figs. (XLSX)

## Acknowledgments

We thank J. Estes and N. Sotuyo for assistance with animal care and M. Osmanski for feedback on behavioral training and the manuscript. The intracellular recording examples were obtained by Dr. Yunyan Wang while a postdoctoral fellow in the Wang Lab.

## Author contributions

**Conceptualization:** Evan D. Remington, Xiaoqin Wang.

**Data curation:** Evan D. Remington.

**Formal analysis:** Chenggang Chen, Evan D. Remington.

**Funding acquisition:** Xiaoqin Wang.

**Investigation:** Chenggang Chen, Evan D. Remington, Xiaoqin Wang.

**Methodology:** Chenggang Chen, Evan D. Remington, Xiaoqin Wang.

**Project administration:** Xiaoqin Wang.

**Resources:** Xiaoqin Wang.

**Software:** Chenggang Chen.

**Visualization:** Chenggang Chen, Evan D. Remington.

**Writing – original draft:** Chenggang Chen, Evan D. Remington.

**Writing – review & editing:** Chenggang Chen, Xiaoqin Wang.

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
