## [Editor Report · Decision Letter 0]

20 Oct 2025

Dear Dr Wang,

Thank you for submitting your manuscript entitled "Dynamic representation of sound locations during task engagement in marmoset auditory cortex" for consideration as a Research Article by PLOS Biology.

Your manuscript has now been evaluated by the PLOS Biology editorial staff as well as by an academic editor with relevant expertise and I am writing to let you know that we would like to send your submission out for external peer review.

Once your full submission is complete, your paper will undergo a series of checks in preparation for peer review. After your manuscript has passed the checks it will be sent out for review. To provide the metadata for your submission, please Login to Editorial Manager (https://www.editorialmanager.com/pbiology) within two working days, i.e. by Oct 22 2025 11:59PM.

Kind regards,

Christian

Christian Schnell, PhD

Senior Editor

PLOS Biology

cschnell@plos.org

---

## [Decision Letter · Decision Letter 1]

9 Dec 2025

Dear Dr Wang,

Thank you for your patience while your manuscript "Dynamic representation of sound locations during task engagement in marmoset auditory cortex" was peer-reviewed at PLOS Biology. It has now been evaluated by the PLOS Biology editors, an Academic Editor with relevant expertise, and by several independent reviewers.

In light of the reviews, which you will find at the end of this email, we would like to invite you to revise the work to thoroughly address the reviewers' reports.

As you will see below, the reviewers overall liked your study. Reviewer 1 has a few suggestions for improving the figures and presentation. Reviewer 2 suggests a number of additional analyses that are required to strengthen the novelty of your study, and mentions a concern regarding the statistical independence that needs to be addressed. Reviewer 3 only lists a few minor concerns regarding the presentation of your findings.

Given the extent of revision needed, we cannot make a decision about publication until we have seen the revised manuscript and your response to the reviewers' comments. Your revised manuscript is likely to be sent for further evaluation by all or a subset of the reviewers.

**IMPORTANT - SUBMITTING YOUR REVISION**

*Re-submission Checklist*

*Published Peer Review*

*PLOS Data Policy*

*Blot and Gel Data Policy*

Sincerely,

Christian

Christian Schnell, PhD

Senior Editor

PLOS Biology

cschnell@plos.org

REVIEWS:

Reviewer #1: General Comments:

This is an awkward report of the effects of behavioral task on spatial receptive fields in early auditory cortical areas in the non-human primate, the marmoset. There are several strengths here, mainly the effects of the behavioral task/attention on these neural responses, and the large spatial region (very close to spherical) tested, so these data should become part of literature as it will be of interest to many. The primary result is that the behavioral task will influence the response to stimuli in the classically measured receptive field, usually by increases but less commonly by decreases in activity. On the down side, figures 1 and 2 could use some serious work, and while supportive, figures 3 - 5 make only minor contributions to the overall impact of the study, but are still worthwhile in the grand scheme of things. Fortunately, all these are salvageable but will take some effort on the author's part. The paper is also fraught with mis-statements, grammatical errors, etc., of which this reviewer gave up about half way through in noting in 'minor' comments. A serious re-view and re-editing of the entire manuscript with an eye to the details will make the manuscript much less distracting

Major:

"Dynamic spatial responses, in

146 particular outside RFs, could support functions such as auditory streaming and the cocktail-party

147 effect, and possibly account for the absence of an ordered cortical map of auditory space

148 (Middlebrooks, 2021)."

There is no support for this statement in the results provided and it should be omitted.

Figure 1 provides more confusion than clarity and the reference to this figure in the text is inaccurate. One work around is to show 1D, the behavioral paradigm, and improve the apparatus figurine to be 3D and more accurately reflect the speaker locations relative the a (stylized) monkey (something similar to the Middlebrooks paper). This could be placed near the beginning of the results and referred to there while directing the reader to the Methods section for more details.

Figure 2 is a bit confusing as it is not logically arranged in different panels. Splitting it into two different figures or re-arranging the panels would probably help.

Figure 2: panels E and H are very compelling. MI for these two cells at these locations should be provided so that the ready gets some indication of what the MI means.

Line 270: should probably read "profiles"

Minor:

Line 73 "azimuth and horizontal axes" makes no sense. Is it supposed to be "azimuth and elevation axes"?

The internal dimensions of the sound booth should be provided.

Presumably, lines 126 and 128 are the questions referred to in line 137 as "these questions". It would be easier flow if those lines were put at the end of the paragraph instead of the middle.

Gao et al. 2015 is not in the reference list

Line 613 states that the behavioral paradigm is in Fig 1A but more likely refers to Fig 1D.

Line 634 refers to Fig 1B and C but more likely mean Fig 1E and F.

Line 164: should refer that this is a single unit, not 'the receptive field'

Figure 2 legend: should stat that it is the receptive field of a single neuron, not of a monkey.

Figure 2 legend: C states that the vertical bar represents 200 ms which makes little sense. Is the y-axis representing 200 spikes? Spkes/sec? spikes/stimulus? Is there supposed to be a horizontal bar that represents 200 ms? Or is it the vertical shaded bar as opposed to a line?

Line 316 should read: "… models were first …"

Reviewer #2: This study recorded single neurons in marmoset auditory cortex while animals performed a spatial Go/No-Go task that required detecting changes in sound location. By comparing behavior and passive listening, the authors show that many neurons exhibit large firing-rate increases at target locations, including locations outside their classical passive receptive fields, effectively creating "temporary" receptive fields during behavior. These effects were strongest in caudal auditory areas, and additional passive intracellular and random spatial-profile recordings revealed widespread suppressive inputs that may underlie this dynamic modulation. A divisive‐normalization model is proposed to explain how behavior selectively enhances responses at behaviorally relevant spatial locations. It is a potentially interesting study on dynamic, behavior-dependent modulation of spatial tuning across multiple auditory cortical areas. The manuscript is carefully crafted and analytically rigorous, integrating a demanding behavioral paradigm with full-field spatial receptive-field mapping in marmoset auditory cortex. The analyses are thoughtfully executed, and the dataset is technically solid and comprehensive.

My suggestions to strengthen the work:

1. The manuscript would benefit from a clearer articulation of its conceptual advance, as the current framing may feel incremental relative to prior work; more explicitly highlighting what fundamentally new insight about dynamic spatial coding emerges from these results would help strengthen the broader impact of the study.

2. The interpretation of the observed modulation as "attention" should be clarified. Because a Go/No-Go task inherently engages arousal, reward expectation, motor preparation, and temporal predictability, it is important to more clearly distinguish these factors or explicitly acknowledge their potential contributions to the neural effects.

3. The link between neural activity and behavior could be strengthened by relating modulation strength more directly to performance measures, e.g. by comparing hit versus miss trials or correlating modulation indices with session-level behavioral sensitivity, to demonstrate that the observed neural changes are behaviorally meaningful.

4. Statistical independence concerns should be addressed, as treating each spatial location as an independent data point may inflate significance. Incorporating unit-level summary metrics or mixed-effects approaches would provide more robust population-level conclusions.

5. The boundaries between newly collected behavioral data and previously published passive or intracellular recordings could be more clearly delineated, especially when these datasets are used to support mechanistic interpretations, to avoid confusion about what was measured in behaving animals versus passive conditions.

6. The normalization model, while plausible, remains largely descriptive; constraining it more tightly with empirical data, for example by fitting parameters to reproduce the observed modulation patterns, would make the mechanistic claims more convincing.

Reviewer #3: In this study, Chen and colleagues investigated how attention to sound location affects neural activity in the auditory cortex. To this end, marmosets were trained to attend to specific sound locations. The authors found that while most neurons increased their firing rates for target sounds presented inside their RFs, many also showed enhanced responses to sounds outside their RFs, sometimes even exceeding responses at the RF center during passive listening. This off-RF facilitation was especially prominent in the caudal auditory cortex. A computational model explained this effect through widespread neural suppression, which was confirmed by physiological evidence of reduced firing rates and hyperpolarized membrane potentials obtained from intracellular recordings. Overall, the results indicate that behavioral demands expand the neural population involved in processing spatial sound beyond those normally active during passive listening.

The study is well written, methodologically sound, and the number of animals used is appropriate for research involving non-human primates. The results are convincing and clearly discussed. Overall, the study appears ready for publication, with only a few minor comments. This work will be of great interest and is well suited for publication in PLOS Biology.

Minor comments:

Figure 1: Please clearly indicate which subfigures illustrate behavioral tasks from previous studies and which are from the current study.

Figure 2: The authors may want to consider using different colors, as the current ones are difficult to distinguish from each other.

---

## [Decision Letter · Decision Letter 2]

19 Feb 2026

Dear Dr Wang,

Thank you for your patience while we considered your revised manuscript entitled "Dynamic representation of sound locations during task engagement in marmoset auditory cortex" for publication as a Research Article at PLOS Biology. This revised version of your manuscript has been evaluated by the PLOS Biology editors, the Academic Editor and one of the original reviewers.

Based on the reviews, we are likely to accept this manuscript for publication, provided you satisfactorily address the data and other policy-related requests stated below my signature.

In addition, we would like you to consider a suggestion to improve the title:

"Behavioral engagement facilitates auditory neuron responses beyond their receptive fields"

We expect to receive your revised manuscript within two weeks.

*Published Peer Review History*

*Press*

Sincerely,

Ines

--

Ines Alvarez-Garcia, PhD

Senior Editor

PLOS Biology

on behalf of

Christian Schnell, PhD

Senior Editor

PLOS Biology

cschnell@plos.org

ETHICS STATEMENT:

-- Please include an approval number in your ethics statement.

DATA POLICY:

Fig. 1D-F; Fig. 2A-D; Fig. 3A-F; Fig. 4A-D; Fig. 5B-E; Fig. 7A-C; Fig. S1A-F; Fig. S2A, B; Fig. S3A-C; Fig. S4A-D and Fig. S5A-D

Please also ensure that figure legends in your manuscript include information on WHERE THE UNDERLYING DATA CAN BE FOUND. For the data deposited in GitHub, please add the link in all the corresponding figure legends.

***We noted that you have deposited some data in GitHub, but given the format of the files, we cannot access the data, so please make sure you include all the data needed according to our policy and clearly indicate what data belongs to each figure and sections to help the reader.

***Please make sure that you have the right permission from OUP to reuse the data shown in Fig. 6 from your article published in Cereb Cortex 29, 1199-1216 (2019) https://academic.oup.com/pages/standard-publication-reuse-rights

CODE POLICY

Reviewers' comments

Rev. 2:

My points are well addressed.

---

## [Editor Report · Decision Letter 3]

27 Feb 2026

Dear Dr Wang,

Thank you for the submission of your revised Research Article "Behavioral engagement facilitates auditory neuron responses beyond their receptive fields" for publication in PLOS Biology. On behalf of my colleagues and the Academic Editor, Manuel Malmierca, I am pleased to say that we can in principle accept your manuscript for publication, provided you address any remaining formatting and reporting issues. These will be detailed in an email you should receive within 2-3 business days from our colleagues in the journal operations team; no action is required from you until then. Please note that we will not be able to formally accept your manuscript and schedule it for publication until you have completed any requested changes.

PRESS

Sincerely,

Christian

Christian Schnell, PhD

Senior Editor

PLOS Biology

cschnell@plos.org